# Domain-adaptive neural networks improve supervised machine learning based on simulated population genetic data

Ziyi Mo[1,2], Adam Siepel[1,2]*

1 Simons Center for Quantitative Biology, Cold Spring Harbor Laboratory, Cold Spring Harbor, New York, United States of America, 2 School of Biological Sciences, Cold Spring Harbor Laboratory, Cold Spring Harbor, New York, United States of America

* asiepel@cshl.edu

**Data Availability Statement:** All code used in this study are available at github.com/ziyimo/popgen-dom-adapt. The 1000 Genomes data are available at www.internationalgenome.org/data.

## Abstract

Investigators have recently introduced powerful methods for population genetic inference that rely on supervised machine learning from simulated data. Despite their performance advantages, these methods can fail when the simulated training data does not adequately resemble data from the real world. Here, we show that this "simulation mis-specification" problem can be framed as a "domain adaptation" problem, where a model learned from one data distribution is applied to a dataset drawn from a different distribution. By applying an established domain-adaptation technique based on a gradient reversal layer (GRL), originally introduced for image classification, we show that the effects of simulation mis-specification can be substantially mitigated. We focus our analysis on two state-of-the-art deep-learning population genetic methods—SIA, which infers positive selection from features of the ancestral recombination graph (ARG), and ReLERNN, which infers recombination rates from genotype matrices. In the case of SIA, the domain adaptive framework also compensates for ARG inference error. Using the **d**omain-**ada**ptive SIA (dadaSIA) model, we estimate improved selection coefficients at selected loci in the 1000 Genomes CEU population. We anticipate that domain adaptation will prove to be widely applicable in the growing use of supervised machine learning in population genetics.

## Author summary

Population genetic simulation is a powerful tool in the study of evolution. A number of supervised machine learning methods have been developed that take advantage of inexpensive simulations as training data. Despite their outstanding performance in benchmarks, these models can fail when the simulated training data deviate from the real data. In this work, we employed domain adaptation techniques to address this "simulation mis-specification" problem by training the machine learning model jointly with simulated and real data. We performed extensive benchmark experiments to demonstrate the improvement of the domain-adaptive models over standard machine learning models in the presence of different types of mis-specification. In addition, we applied *dadaSIA*, a domain-

**Funding:** This research was supported by US National Institutes of Health grant R35-GM127070 (to AS), the CSHL School of Biological Sciences Gladys & Roland Harriman Fellowship (to ZM), and the Simons Center for Quantitative Biology at Cold Spring Harbor Laboratory. The content is solely the responsibility of the authors and does not necessarily represent the official views of the US National Institutes of Health. The funders had no role in study design, data collection and analysis, decision to publish, or preparation of the manuscript.

**Competing interests:** The authors have declared that no competing interests exist.

adaptive selection inference model, to improve the estimates of selection coefficients at selected loci in a European population. The domain adaptation framework proposed in our work is widely applicable to models relying on synthetic training data and therefore opens the door to many more applications in population genetics.

## Introduction

Advances in genome sequencing have allowed population genetic analyses to be applied to many thousands of individual genome sequences [1–3]. Given adequately rigorous and scalable computational tools for analysis, these rich catalogs of genetic variation provide opportunities for addressing many important questions in areas such as human evolution, plant genetics, and the ecology of non-model organisms. Deep-learning methods, already well-established in other application areas [4], have proven to be good matches for these analytical tasks and have recently been successfully applied to many problems in population genetics [5–14].

The key to the success of deep learning in population genetics has been the use of large amounts of simulated data for training. Under simplifying, yet largely realistic, assumptions, evolution plays by relatively straightforward rules. By exploiting these rules and advances in computing power, a new generation of computational simulators has made it possible to efficiently produce large quantities of perfectly labeled synthetic data across a wide range of evolutionary scenarios [15–17]. At the same time, programming libraries such as stdpopsim have made these simulators accessible to a broad community of researchers while improving the reproducibility of simulation workflows [18,19]. The facility of generating synthetic training data serves as the foundation of the new simulate-and-train paradigm of supervised machine learning for population genetics inference (**Fig 1A**; [7,13]).

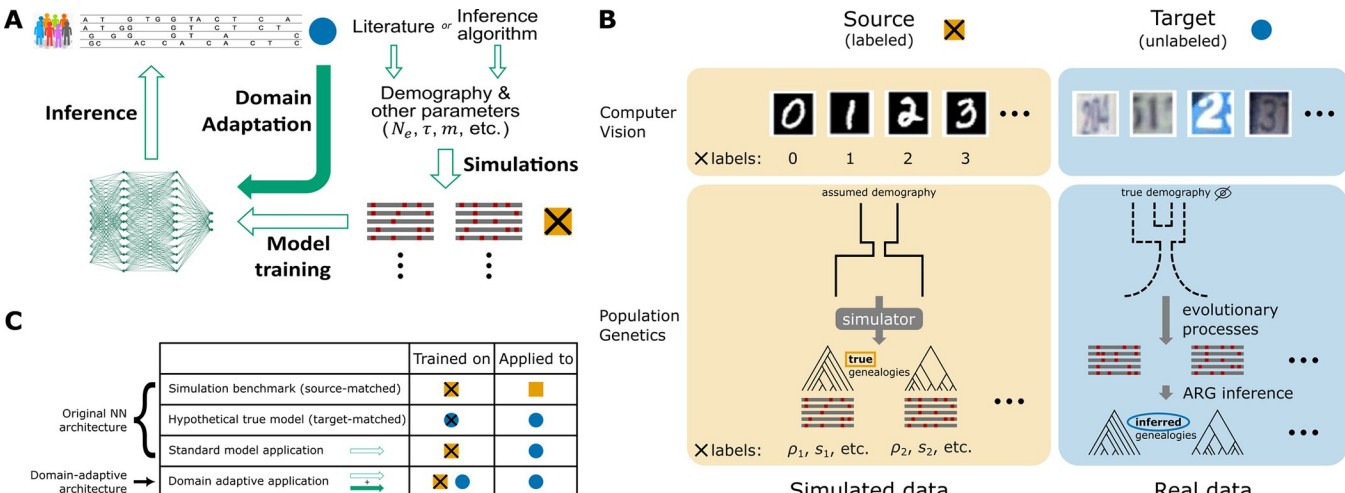

**Fig 1. Unsupervised domain adaptation in the context of population genetic inference. A)** A high-level overview of the supervised machine-learning approach for population genetic inference and how domain adaptation fits into the paradigm. **B)** Example formulations of the unsupervised domain adaptation problem with application to computer vision and population genetics. Note that in the specific case of SIA, which uses features of the ARG, the source domain data always consist of *true* genealogies generated in simulations, whereas the target domain data always consist of *inferred* genealogies reconstructed from observed sequence data. **C)** Four benchmarking scenarios considered in this study. The original model was both trained and tested on source domain data (simulation benchmark), both trained and tested on target domain data (hypothetical true model), or trained on source domain data but applied to target domain data (standard model application). These three cases contextualize the performance of the domain-adaptive model (see **Methods** for details). Gold squares represent source domain data, blue circles represent target domain data and crosses (**x**) represent labels.

At the same time, this paradigm is highly dependent on well-specified models for simulation [13]. If the simulation assumptions do not match the underlying generative process of the real data—that is, in the presence of *simulation mis-specification*—the trained deep-learning model may reflect the biases in the simulated data and perform poorly on real data. Indeed, previous studies have shown that, despite being robust to mild to moderate levels of mis-specification, performance inevitably degrades when the mismatch becomes severe [10,12].

In a typical workflow, key simulation parameters such as the mutation rate, recombination rate, and parameters of the demographic model are either estimated from the data or obtained from the literature (**Fig 1A**; [18,19]). Sometimes these parameters are allowed to vary during simulation, and sometimes investigators evaluate the sensitivity of predictions to departures from the assumed range, but there is typically no way to ensure that the ranges considered are adequately large. Moreover, these benchmarks do not usually account for under-parameterization of the demographic model. Particularly in the case of non-model organisms, the quality of the estimates can be further limited by the availability of data. Overall, some degree of mis-specification in the simulated training data is impossible to avoid.

One way to mitigate the effects of simulation mis-specification would be to engineer a simulator to force the simulated data to be compatible with real data. For example, one could simulate from an overdispersed distribution of parameters followed by a rejection sampling step (based on summary statistics) as in Approximate Bayesian Computation (ABC) methods, or one could use a Generative Adversarial Network (GAN) [20] to mimic the real data. These methods tend to be costly, however. For example, ABC methods scale poorly with the dimensionality of the parameter space, and GANs are notoriously hard to train.

Here we consider the alternative approach of adopting a deep-learning model that is explicitly designed to account for and mitigate the mismatch between simulated and real data (**Fig 1A**). A standard machine learning model aims to make accurate predictions on data following the same probability distribution as the training instances. In contrast, the task of building well-performing models for a target dataset that has a *different* distribution from the training dataset is termed "domain adaptation" in the machine-learning literature [21,22]. A typical setting of interest for domain adaptation is image classification (**Fig 1B**). For example, suppose a digit-recognition model is needed for the Street View House Numbers (SVHN) dataset (the "target domain"), but abundant labeled training data is only available from the MNIST dataset of handwritten digits (the "source domain"). In this case, a method needs to train on one dataset and perform well on another, despite systematic differences between the two data distributions.

Various strategies for domain adaptation have been introduced. Prior to the advent of deep learning, early methods focused on reweighting training instances according to their likelihoods of being a source or target example [23,24] or explicitly manipulating a feature space through augmentation [25], alignment [26,27] or transformation [28]. Recently, specialized neural network architectures have been developed for deep domain adaptation. Most model architectures of this kind share the common goal of learning a "domain-invariant" representation of the data through a feature extractor neural network, for example, by minimizing domain divergence [29], by adversarial training [30,31] or through an auxiliary reconstruction task [32]. Domain adaptation so far has been most widely applied in the fields of computer vision (e.g., using stock photos for semantic segmentation of real photos) and natural language processing (e.g., using Amazon product reviews for sentiment analysis of movies and TV shows) where large, heterogeneous datasets are common but producing labeled training examples can be labor intensive [22]. More recently, deep domain adaptation has been used in regulatory genomics to enable cross-species transcription-factor-binding-site prediction [33].

In this work, we reframe the simulation mis-specification problem in population genetics as an unsupervised domain adaptation problem—unsupervised in the sense that data from the target domain is not labeled (**Fig 1B**). In particular, we use population-genetic simulations to obtain large amounts of perfectly labeled training data in the source domain. We then seek to apply the trained model to unlabeled real data in the target domain. We use domain adaptation techniques to explicitly account for the mismatch between these two domains when training the model.

To demonstrate the feasibility of this approach, we incorporated a domain-adaptive neural network architecture into two published deep learning models for population genetic inference: 1) SIA [12], which identifies selective sweeps based on the Ancestral Recombination Graph (ARG), and 2) ReLERNN [10], which infers recombination rates from raw genotypic data. Through extensive simulation studies, we demonstrated that the domain adaptive versions of the models significantly outperformed the standard versions under realistic scenarios of simulation mis-specification. Our domain-adaptive framework for utilizing mis-specified synthetic data for supervised learning opens the door to many more applications in population genetics.

## Results

### Experimental design

We created domain-adaptive versions of the SIA and ReLERNN models, each of which employed a gradient reversal layer (GRL) [30] (**Fig 2A and 2B**). As noted, the goal of domain adaptation is to establish a "domain-invariant" representation of the data (**Fig 1A**). Our neural networks consist of two major components: the original networks ("feature extractor" in green and "label predictor" in blue in **Fig 2A and 2B**), which are applied only to labeled examples from the "source" (simulated) domain; and alternative branches ("domain classifier" in yellow in **Fig 2A and 2B**), which use the same feature-extraction portions of the first networks but have the distinct goal of distinguishing data from the "source" (simulated) and "target" (real) domains (they are applied to both). When the neural network is trained by back-propagation,

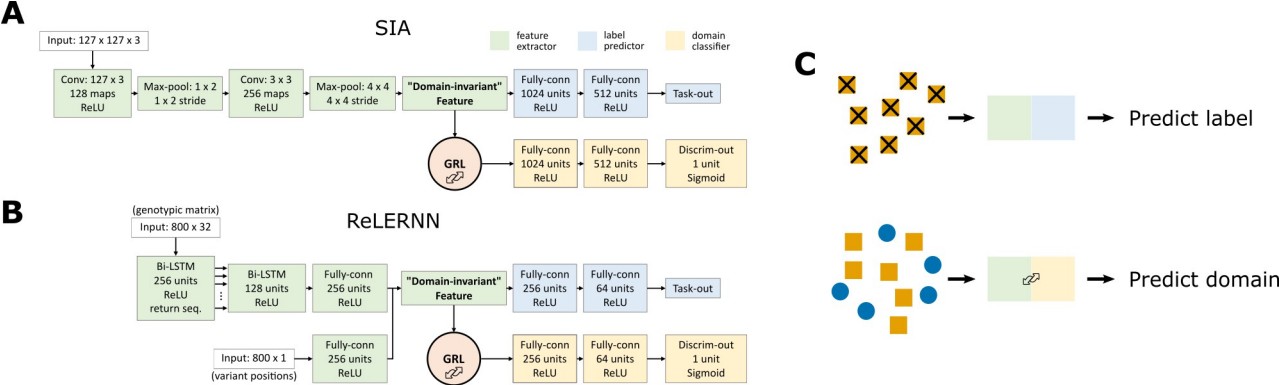

**Fig 2. Neural network architecture for domain adaptation.** The model architectures incorporating gradient reversal layers (GRLs) for **A)** SIA and **B)** ReLERNN. The feature extractor of SIA contains $1.49 \times 10^5$ trainable parameters, whereas the label predictor and domain classifier contains $1.22 \times 10^8$ each. The feature extractor of ReLERNN contains $1.52 \times 10^6$ trainable parameters, whereas the label predictor and domain classifier contains $1.49 \times 10^5$ each. Note that the total number of trainable parameters includes those in batch normalization layers. **C)** When training the networks, each minibatch of training data consists of two components: (1) labeled data from the source domain fed through the feature extractor and the label predictor; and (2) a mixture of unlabeled data from both the source and target domains fed through the feature extractor and the domain classifier. The first component trains the model to perform its designated task. However, the GRL inverts the loss function for the second component, discouraging the model from differentiating the two domains and leading to the extraction of "domain-invariant" features.

the GRL reverses the sign of the gradient for the feature extractor with respect to the domain-classifier loss. By doing so, the GRL systematically undermines this secondary goal of distinguishing the two domains (**Fig 2**, see **Methods** for details), and therefore promotes domain invariance in feature extraction.

We designed two sets of benchmark experiments to assess the performance of the domain-adaptive models relative to the standard models. In both cases, we tested the methods using "real" data in the target domain that was actually generated by simulation, but included features not considered by the simpler simulator used for the source domain. In the first set of experiments, background selection was present in the target domain but not the source domain. In the second set of experiments, the demographic model used for the source-domain simulations was estimated from "real" data generated under a more complex demographic model and was therefore somewhat mis-specified (as detailed below). Below we refer to these as the "background selection" and "demography mis-specification" experiments.

## Performance of domain-adaptive SIA model

We compared the performance of the **d**omain-**ada**ptive SIA (dadaSIA) model to that of the standard SIA model on held-out "real" data, considering both a classification (distinguishing selective sweeps from neutrality) and a regression (inferring selection coefficients) task. In all cases, we focused on a comparison of the domain-adaptive model to the standard case where a model is simply trained on data from the source domain and then applied to the target domain ("standard model"; **Fig 1C**). Note that the version of SIA used by both the domain-adaptive and standard models includes a variety of minor improvements that led to modest gains in performance over the previously published version (see *Updates to genealogical features and deep learning architecture for the SIA model* in **Methods** and **S1B and S1C Fig**). The codebase of the original SIA model has been updated accordingly.

For additional context, we also considered the two cases where the training and testing domains matched ("source-matched" or "target-matched"; **Fig 1C**)—although we note that these cases are not achievable with real data and provide only hypothetical upper bounds on performance. Notably, in the source-matched (or "simulation benchmark") case, the standard model is both trained and tested with true genealogies from source-domain simulations. By contrast, in the target-matched (or "hypothetical true model") case, the standard model is trained as if target-domain data with ground-truth selection coefficient labels were available. Since genealogies need to be inferred in the target domain (**Fig 1B**), the hypothetical true model is both trained and tested with inferred genealogies (see *Setup of benchmarking experiments* in **Methods** for details).

As noted, we considered two types of mis-specification, background selection and demographic mis-specification. In the background selection experiments, the target domain experienced selection in a central "genic" region (following a DFE from [34]), leading to background selection in flanking regions. This genic region was omitted in the source domain. In the demographic mis-specification experiments, the demographic model for source-domain simulations was inferred from "real" data using G-PhoCS [35]. Both the real (target domain) and inferred (source domain) models assumed three populations with migration, but the inferred model was under-parameterized and its parameters differed substantially from the real model (**S1A Fig**) (see **Methods** for details).

In both the background selection and demography mis-specification experiments, and in both the classification and regression tasks, the domain-adaptive SIA model substantially improved on the standard model (**Fig 3**). Indeed, in all cases, the domain-adaptive model (turquoise lines in **Fig 3A and 3C**) nearly achieved the upper bound of the hypothetical true

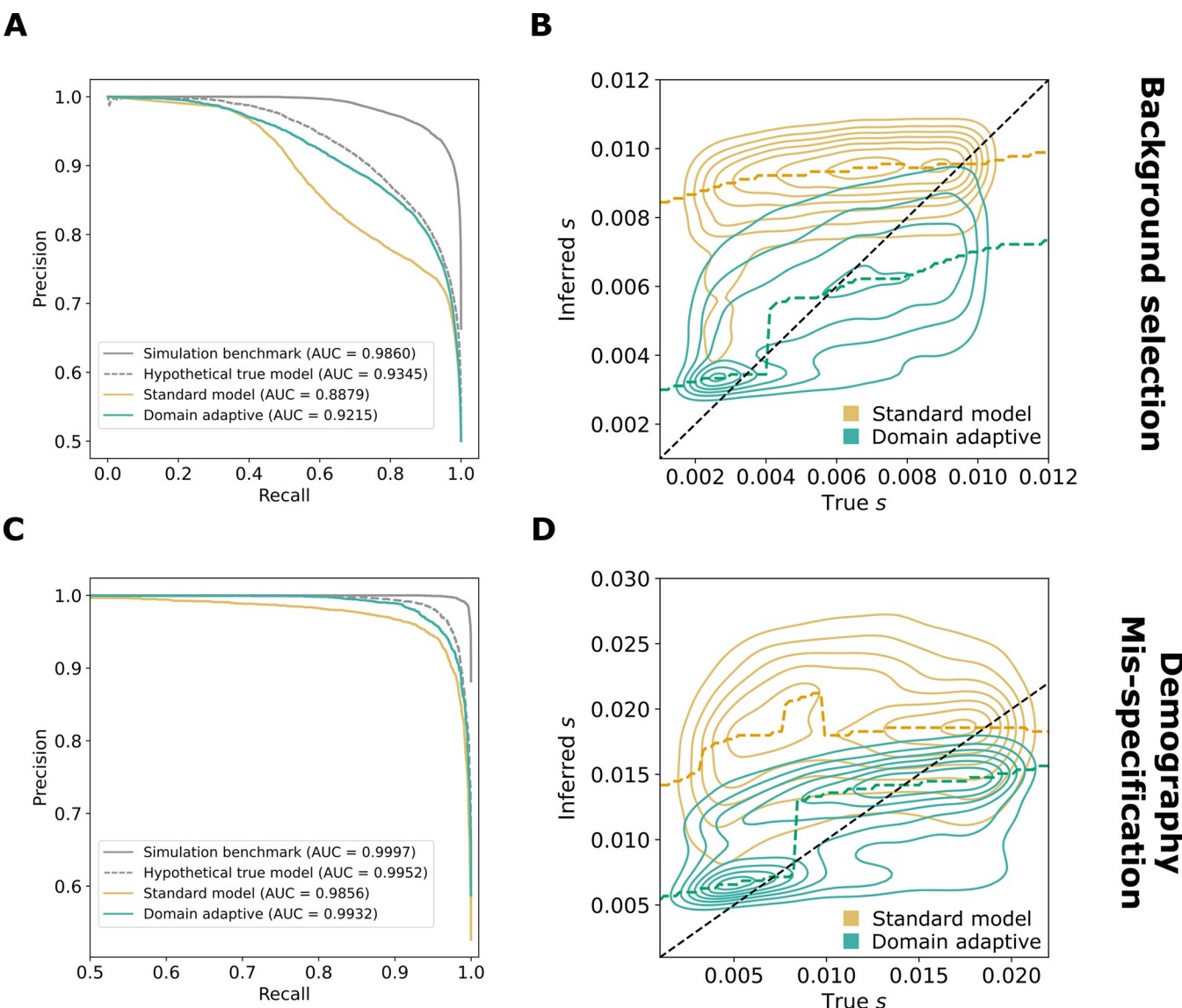

**Fig 3. Performance of domain-adaptive SIA models.** Results are shown from (**A, B**) the background-selection and (**C, D**) the demography-mis-specification experiments. (**A, C**) Precision-recall curves for sweep classification. (**B, D**) Contour plots summarizing true (horizontal axis) vs. inferred (vertical axis) selection coefficients (*s*) for the standard (gold) and domain adaptive (turquoise) models as evaluated on the held-out test dataset. The ridge along the horizontal axis of each contour is traced by a dashed line, representing the mode of the inferred value for each true value of *s*. Raw data underlying the contour plots are presented in **S2 Fig**. See **Fig 1C** for definition of the model labels.

model (dashed gray lines) and clearly outperformed the standard model (gold lines), suggesting that domain adaptation had largely "rescued" SIA from the effects of simulation mis-specification (see also **S2C and S2D Fig**). The standard model performed particularly poorly on the regression task (**Fig 3B and 3D**), but the domain-adaptive model achieved substantial improvements, reducing both the absolute error as well as the upward bias of the estimation (**S2C and S2D Fig**).

The comparisons with the simulation benchmark and hypothetical true model were also informative in other ways. Notice that performance in the simulation benchmark case was considerably better than that in all other cases, including the hypothetical true model. For SIA

in particular, the ARG is "known" (fixed in simulation) in the source domain, whereas in the target domain it must be inferred (**Fig 1B**). Thus, the difference between the simulation benchmark (source-matched) and hypothetical true model (target-matched) cases represents a rough measure of the importance of ARG inference error (see **Discussion**). In addition, note that in many studies, benchmarking of population-genetic models is performed using the same, or similar, simulations as those used for training, as with our hypothetical true model. Thus, the difference between the hypothetical true model and the standard model is representative of the degree to which benchmarks of this kind may be overly optimistic about performance, depending on the degree to which the simulations are mis-specified.

We further investigated the effect of imbalanced training data from the target domain on the performance of the domain-adaptive model in the context of sweep classification. Despite the ability to simulate perfectly class-balanced labeled data in the source domain, in practice we have no control over whether real data are balanced. Using simulations for the background selection mis-specification experiments, we tested the performance of the domain-adaptive SIA model classifying sweeps when trained with unlabeled "real" data under different proportions of sweep vs. neutral examples. While a balanced dataset yielded the best performance, significantly skewed datasets (20% or 80% sweep examples) still provided the domain-adaptive model with reasonable improvement upon the standard model (**S3A and S3B Fig**). The exception appeared to be when the target domain data consisted entirely of sweep examples (100% sweep). Although highly unrealistic, this scenario demonstrates that the domain-adaptive model can underperform the standard model when the target domain data follow a radically different distribution.

Another type of imbalance arises if only a limited amount of target domain data is available to train the domain-adaptive model. Using the same set of simulations for the background selection mis-specification experiments, we tested the performance of the domain-adaptive SIA model when trained with less target domain data. With the target domain data at only 10% of the source domain data (source:target ratio = 10:1), the model suffered a noticeable drop in performance yet still maintained a clear advantage over the standard model (**S3C and S3E Fig**). We did not examine the case where there is more target domain than source domain data, since one could always simulate additional source domain data to match the size of the target domain. In summary, our experiments suggest that domain adaptation can accommodate reduced or imbalanced data for the target domain but there is a cost in performance if the reduction or imbalance is extreme.

## Performance of domain-adaptive ReLERNN model

We performed a parallel set of experiments with a domain-adaptive version of ReLERNN. In this case, the background selection experiment was essentially the same as for SIA, but we used a simpler design for the demography mis-specification experiment, following [10]. Briefly, the "real" (target domain) data was generated according to the out-of-Africa European demographic model estimated by [36]. By contrast, the simulated data for the source domain simply assumed a constant-sized panmictic population at equilibrium with $N_e = \frac{\hat{\theta}_W}{4\mu}$, where $\hat{\theta}_W$ is the Watterson estimator obtained from the "real" data (see **Methods** for details).

Similar to our results for SIA, the domain-adaptive ReLERNN model both reduced the mean absolute error (MAE) and corrected for the downward bias in recombination-rate estimates compared to the standard model (**Figs 4 and S4**). In the background-selection experiment, the standard ReLERNN model performed quite well (**Figs 4A and S4A,** MAE = $5.60 \times 10^{-9}$), but the domain-adaptive ReLERNN model nonetheless further reduced the MAE to $4.41 \times 10^{-9}$ (**S4C Fig**, Welch's $t$-test: $n$ = 25,000, $t$ =31.0, $p < 10^{-208}$). The advantage

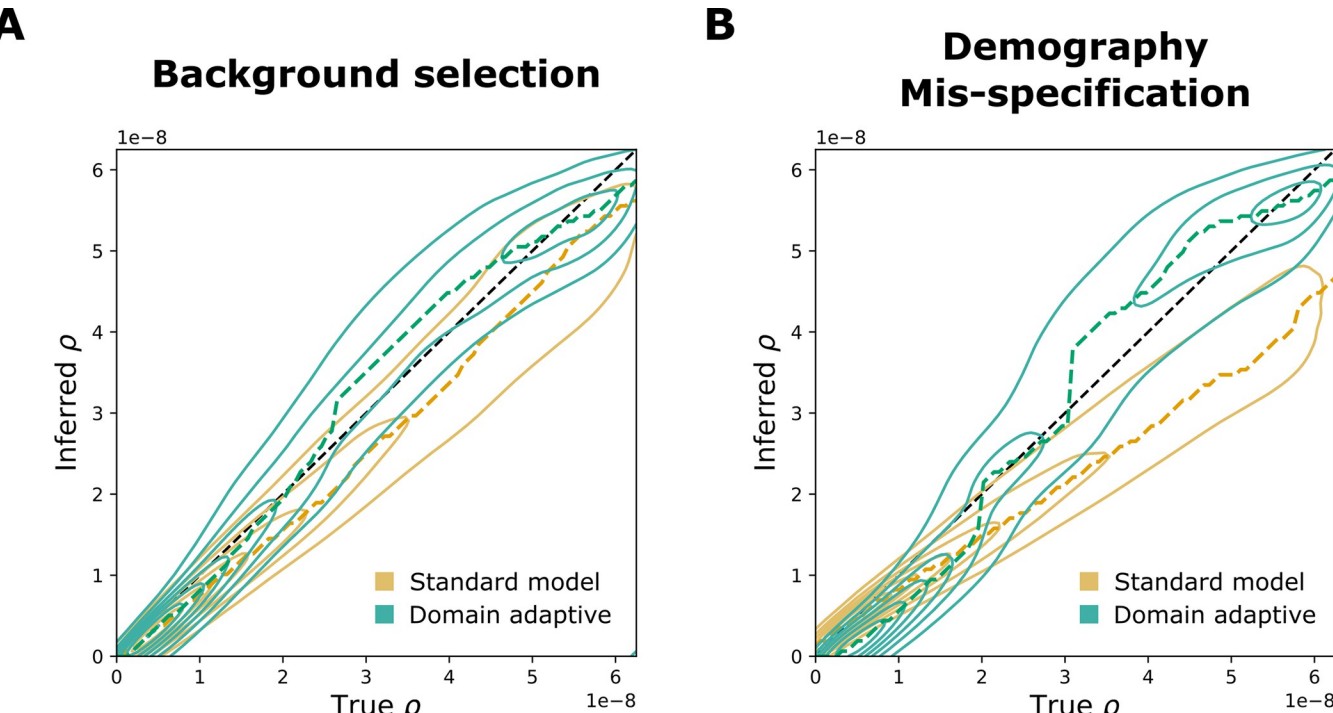

**Fig 4. Performance of domain-adaptive ReLERNN models.** Results are shown from (**A**) the background-selection and (**B**) the demography-mis-specification experiments. Each contour plot summarizes true (horizontal axis) vs. inferred (vertical axis) recombination rates ($\rho$) for the standard (gold) and domain adaptive (turquoise) models as evaluated on the held-out test dataset. The ridge along the horizontal axis of each contour is traced by a dashed line, representing the mode of the inferred value for each true value of $\rho$. Raw data underlying the contour plots are presented in **S4 Fig**.

of the domain-adaptive model was more apparent in the demography-mis-specification experiment (**Figs 4B** and **S4B**), where it reduced the MAE from $8.06 \times 10^{-9}$ to $5.45 \times 10^{-9}$ (**S4D**, Welch's t-test, $n = 25{,}000$, $t = 72.4$, $p < 10^{-323}$). Notably, our results for the standard model in the demography-mis-specification experiment were highly similar to those reported by [10], including the approximate mean and range of the raw error (compare Fig 4A from [10] and **S4D Fig**), as well as the downward bias.

Interestingly, Adrion et al. [10] observed that ReLERNN was sometimes more strongly influenced by demographic mis-specification than unsupervised methods such as LDhelmet, even though it still performed better in terms of absolute error. The addition of domain adaptation appears to considerably mitigate this susceptibility to demographic mis-specification, making an excellent method even stronger.

### Efficacy of domain adaptation under various degrees of simulation mis-specification

So far, we have examined scenarios of relatively modest simulation mis-specification, likely to be encountered in real applications. While domain adaptation appeared to be effective in these cases, we expect a limit to its capability when mis-specification is extreme. We therefore carried out a series of experiments to probe the performance of the dadaSIA model under increasingly severe simulation mis-specification (**S4 Fig**, also see **Methods**).

We found that dadaSIA exhibited good performance when mis-specification was caused by genealogy inference alone or by light to moderate bottlenecks. As the bottleneck became more severe, its performance deteriorated, but even with a 5% bottleneck, dadaSIA still

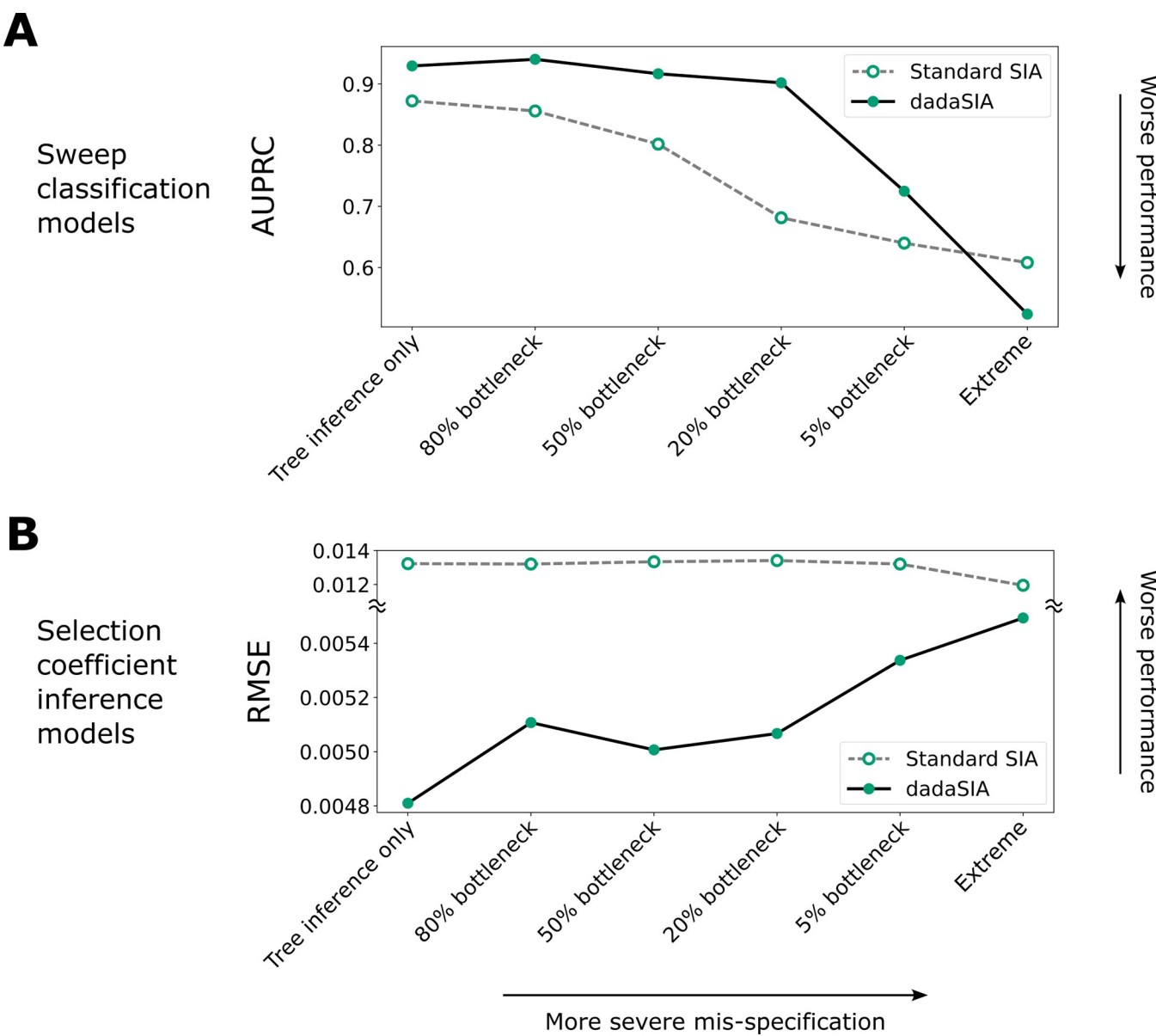

**Fig 5. Performance of domain-adaptive SIA (dadaSIA) model with different degrees of mis-specification.** The performance of the model on the sweep classification task is quantified by the area under the precision-recall curve (AUPRC) (**A**). Performance on the selection-coefficient inference task is quantified by root mean squared error (RMSE) (**B**). In the "tree inference only" case, there is no mis-specification other than that caused by error in genealogy inference. In the "extreme" case, mis-specification consists of a 5% bottleneck, background selection and an 8-fold mis-specification in recombination rate. See **S4 Fig** for illustrations of the different bottlenecks and **Methods** for details.

outperformed the standard model (**Fig 5**). To examine the limits of the method, we tested an extreme scenario with the 5% bottleneck, background selection and an 8-fold mis-specification of recombination rate. In this case, the model performed poorly, having virtually no power to classify sweeps and large errors in its selection coefficient estimates (**Fig 5**). This example demonstrates that, while domain adaptation is useful over a broad range of mis-specification levels, it eventually does fail when mis-specification becomes extreme.

Does domain adaptation compromise performance at the opposite extreme, where there is little or no simulation mis-specification? To address this question, we tested the standard and

domain-adaptive ReLERNN models in a setting without any simulation mis-specification. We focused here on ReLERNN, which directly uses raw genotypic data, as opposed to SIA, which always has some mis-specification due to genealogy inference error. We observed that the standard and domain-adaptive ReLERNN models performed nearly identically when no mis-specification was present, with only minor decreases in performance (**S7 Fig**). Thus, there is perhaps some cost in using domain adaptation when it is not needed, but, at least in our case, that cost appears to be slight.

### Application of domain-adaptive SIA to real data

In applications to real data, the true selection coefficient is not known, so it is impossible to perform a definitive comparison of methods. Nevertheless, it can be informative to evaluate the degree to which alternative methods are concordant, especially with consideration of their relative performance in simulation studies.

Toward this end, we re-applied our **d**omain-**ada**ptive SIA model (dadaSIA) to several loci in the human genome that we previously analyzed with SIA [12], using whole-genome sequence data from the 1000 Genomes CEU population [1]. For the target domain, we sampled genealogies from genome-wide ARGs inferred from the individual sequences (see **Methods**). The putative causal loci analyzed included single nucleotide polymorphisms (SNPs) at the *LCT* gene [37], one of the best-studied cases of selective sweeps in the human genome; at the disease-associated genes *TCF7L2* [38], *ANKK1* [39] and *FTO* [40]; at the pigmentation genes *KITLG* [41], *ASIP* [42], *TYR* [41,42], *OCA2* [43,44], *TYRP1* [45] and *TTC3* [46], which were also analyzed by [47]; and at the genes *MC1R* [41,43] and *ABCC11* [48], where SIA reported novel signals of selection.

We found that dadaSIA generally made similar predictions to SIA at these SNPs, but there were some notable differences. The seven loci predicted by SIA to be sweeps were also predicted by dadaSIA to be sweeps (**Table 1**), although dadaSIA always reported higher confidence in these predictions (with probability of neutrality, $P_{\text{neu}} < 10^{-2}$ in all cases) than did SIA ($P_{\text{neu}}$ up to 0.384 for *TYR*). The five loci predicted by SIA not to be sweeps were also predicted by dadaSIA not to be sweeps ($P_{\text{neu}} > 0.5$). At *LCT*, the strongest sweep considered, the selection coefficient ($s$) estimated by dadaSIA remained very close to SIA's previous estimate of $s = 0.01$ and also close to several prior estimates [37,49,50]. In all other cases, the estimate from SIA was somewhat revised by dada SIA, generally by factors of about 2–3. Importantly, in all cases, the estimates from dadaSIA remained much closer to those from SIA than to estimates by other methods (**Table 1**). Together, these observations suggest that the addition of domain adaptation does not radically alter SIA's predictions for real data but may in some cases improve them (see **Discussion**).

### Discussion

Standard approaches to supervised machine learning rest on the assumption that the data they are used to analyze follow essentially the same distribution as the data used for training. In applications in population genetics, the training data are typically generated by simulation, leading to concerns about potential biases from simulation mis-specification when supervised machine-learning methods are used in place of more traditional summary-statistic- or model-based methods [11,13]. In this article, we have shown that techniques from the "domain adaptation" literature can effectively be used to address this problem. In particular, we showed that the addition of a gradient reversal layer (GRL) to two recently developed deep-learning methods for population genetic analysis—SIA and ReLERNN—led to clear improvements in performance on "real" data that differed in subtle but important ways from the data used to train

**Table 1. Selection coefficients in the European population estimated by domain-adaptive SIA compared to previous estimates.**

| Gene | SNP | Estimates of selection coefficient | | |
|---|---|---|---|---|
| | | Domain-adaptive SIA | SIA* [12] | Previous estimates |
| KITLG | rs12821256 | 0.0035 | 0.0019 | 0.0161 [47] |
| ASIP | rs619865 | 0.0057 | 0.0019 | 0.0974 [47] |
| TYR | rs1393350 | 0.0028 | 0.0011 | 0.0112 [47] |
| OCA2 | rs12913832 | 0.0093 | 0.0056 | 0.002 [47]; 0.036 [62] |
| MC1R | rs1805007 | 0.0027 | 0.0037 | No selection [63] |
| ABCC11 | rs17822931 | 0.0020 | 0.00035 | ~ 0.01 in East Asian [64] |
| LCT | rs4988235 | 0.0097 | 0.010 | ~ 0.01 [37,49,50] |
| TYRP1 | rs13289810 | $P_{neu} > 0.5$ | $P_{neu} > 0.5$ | No selection [47] |
| TTC3 | rs1003719 | $P_{neu} > 0.5$ | $P_{neu} > 0.5$ | No selection [47] |
| TCF7L2 | rs7903146 | $P_{neu} > 0.5$ | $P_{neu} > 0.5$ | N/A |
| ANKK1 | rs1800497 | $P_{neu} > 0.5$ | $P_{neu} > 0.5$ | N/A |
| FTO | rs9939609 | $P_{neu} > 0.5$ | $P_{neu} > 0.5$ | N/A |

* The original SIA model in [12] uses genealogies *inferred* from simulations for training, despite the availability of ground truth genealogies.

the models. These improvements were observed both when the demographic models were mis-specified and when background selection was included in the simulations of "real" data but un-modeled in the training data.

While we observed performance improvements in all of our experiments, they were especially pronounced in the case where SIA was used to predict specific selection coefficients, rather than simply to identify sweeps. The standard model (with training on simulated data and testing on "real" data) performed particularly poorly in this regression setting and domain adaptation produced striking improvements (**Fig 3B and 3D**). This selection-coefficient inference problem appears to be a harder task than either sweep classification or recombination-rate inference, and the performance in this case proves to be more sensitive to simulation mis-specification (cf. **Fig 3A and 3C**). In general, we anticipate considerable differences across population-genetic applications in the value of domain adaptation, with some applications being more sensitive to simulation mis-specification and therefore more apt to benefit from domain adaptation, and others being less so.

We also observed some interesting differences in the ways SIA and ReLERNN responded to domain adaptation. For example, the performance gap between the "simulation benchmark" (trained and tested on simulated data) and "hypothetical true" (trained and tested on real data) models was considerably greater for SIA than for ReLERNN (**S2C, S2D, S4C and S4D Figs**). This difference appears to be driven by ARG inference, which is required by SIA in the hypothetical true case but not the simulation benchmark case, and for which no analog exists for ReLERNN. For SIA, the uncertainty about genealogies given sequence data makes the prediction task fundamentally harder in the real world (target domain) than in simulation (source domain) (**Fig 1B**). By contrast, ReLERNN does not depend on a similar inference task, and therefore the target and source domains are more or less symmetric. This same factor contributed to the much more dramatic drop in performance for SIA than ReLERNN under the "standard model," where the model is trained on simulated data and naively applied to "real" data (**Figs 3B, 3D and 4**). It is, of course, also conceivable that simulation mis-specification has more impact on selection inference than recombination rate inference, rendering the standard SIA model less robust than the standard ReLERNN model. Regardless of the exact cause, the result is more potential for improvement from domain adaptation with SIA than with ReLERNN (**Figs 3, 4, S2 and S4**). In effect, in SIA, domain adaptation not only mitigates

simulation mis-specification but also compensates for ARG inference error, as directly evidenced by the observation that domain adaptation improves model performance when mis-specification is due to genealogy inference alone (**Fig 5**, "Tree inference only"). More broadly, we expect domain adaptation to be especially effective in applications that depend not only on the simulated data itself but also on nontrivial inferences of latent quantities that are known for simulated but not real data.

In addition, we performed a series of experiments to probe the limits of domain adaptation. As expected, the dadaSIA model gradually lost its power as simulation mis-specification became more severe. In an extreme case where mis-specification involved demography, selection and recombination rate, the dadaSIA model had virtually no power to classify sweeps and exhibited high error of selection coefficient inference (**Fig 5**). In practice, simulation models themselves are inferred from real data. With high quality data, state-of-the-art inference tools are unlikely to fail completely (e.g., by missing a 5% bottleneck completely, or under-estimating recombination rate by an order of magnitude). We thus expect the most extreme scenario tested here to be fairly uncommon. Nevertheless, this experiment demonstrated that there are reasonable limits to the efficacy of domain adaptation. Consequently, it is important in real-world applications to begin with the best possible simulation model, before using domain adaptation to further optimize performance.

Because the accuracy of the simulation model is typically not known a priori, it is tempting to apply domain adaptation in all cases, regardless of the true degree of mis-specification. Indeed, we found that the domain-adaptive model performed very similarly to the standard model in the absence of mis-specification (**S7 Fig**), suggesting little risk in applying the approach liberally. When the target domain is mis-specified, the domain classifier appears to "unlearn" the mis-specification, with its loss increasing steadily before plateauing where the source and target domains are no longer distinguishable. In contrast, when there is no mis-specification, the domain classifier starts with a high loss and this loss remains high (**Figs 2B and S8**). In this case, because the source and target domains are effectively indistinguishable, the domain classifier can never do much better than randomly guessing, leading to near-zero gradients along the domain classifier branch. In effect, the training process ignores the domain-classifier branch in this case, and improves only the feature-extractor and label-predictor portions of the model. For this reason, the domain-adaptive model behaves nearly identically to the standard model in the absence of mis-specification.

The accuracy of even the best current selection-coefficient inference methods appears limited [8,9,12,47]. More work is needed on models and methods for inference as well as on the problem of simulation mis-specification. Nevertheless, current methods can still be valuable in approximately characterizing the strength of selection. In our re-analysis of several loci in the 1000 Genomes CEU population, we found that dadaSIA made similar predictions to SIA, but it tended to exhibit higher confidence in its predictions (**Table 1**). Considering the extensive previous work on demography inference for the CEU population, we expect that simulation mis-specification is limited in severity for this analysis, but that some mis-specification is inevitable. Given the similar performance on benchmarks of SIA and other leading methods such as CLUES, their similar sensitivity to moderate levels of simulation mis-specification [12], and the improvements offered by domain adaptation that are demonstrated in this work, we find it likely that dadaSIA improves on previous estimates of selection coefficients in this setting.

In a typical application of domain adaptation, the distribution shift between the source and target domains is treated as a nuisance. However, for certain population genetic questions, the gap between the simulated and real data could in principle help to reveal unmodeled evolutionary processes. We observed that the domain classifier generally tended to start with a lower loss and took more epochs to train when the mis-specification is more severe (**S9 Fig**). It

might be worthwhile, as a future endeavor, to try to identify the features driving this loss, understand their evolutionary significance, and, perhaps, incorporate them into a new set of simulations. In such a way, domain adaptation could be used to discover evolutionary processes and improve the models used for simulation.

Although our experiments were limited to background selection and demographic mis-specification, we expect that the domain adaptation framework would also be effective in addressing many other forms of simulation mis-specification, involving factors such as mutation or recombination rates, or the presence of gene conversion. Another interesting application may be to use domain adaptation to accommodate admixed populations. Each ancestry component could be modeled as a distinct target domain using a multi-target domain adaptation technique [51–53]. It is also worth noting that our experiments considered only one, rather simple, strategy for domain adaptation. Since the GRL was proposed, several other architectures for deep domain adaptation have achieved even better empirical performance on computer vision tasks (see: [54]).

Our domain-adaptation approach leaves simulations unchanged and attempts to "unlearn" their mis-specification, in contrast to other strategies that aim to improve the simulations themselves. For example, the original SIA model was trained with inferred genealogies from the simulated sequences, rather than the true genealogies used to generate the data, to mitigate the effect of genealogy inference error [12]. An alternative approach is to use a GAN to train a simulator that accurately mimics the real data [20]. These methods can require costly preprocessing steps, but they have the advantage of explicitly addressing the simulation mis-specification in an interpretable manner.

It is perhaps worth distinguishing mis-specification *along* the axis of inference—that is, of target parameters such as the selection coefficient—from mis-specification of other "nuisance" parameters (such as demographic parameters), or similarly, other unmodeled aspects of the data-generating process (such as background selection). From our observations, domain adaptation appears to be effective at addressing mis-specification of nuisance parameters or processes, at least if it is not too severe. Mis-specification of the target parameters, however, is clearly a more challenging problem. For example, it seems unlikely that domain adaptation will ever be able to "extrapolate" beyond the range of the training examples (as it fails to do in **S5 Fig**). Hence, it is essential in practical applications to simulate the parameter of interest from an adequately large range. Notably, Burger et al. [55] recently developed a method that addresses mis-specification in the distribution (but not the range) of a target parameter. Their method improves inference of the scaled mutation rate when regions of the parameter space are under-sampled in the training simulations by adaptively reweighing the training data, effectively improving interpolation (but not extrapolation) from the training distribution. We view these interrelated questions of how to accommodate mis-specification of both nuisance and target parameters as promising areas for future work.

Mis-specification is not only a problem in the simulation-based supervised machine learning setting explored in this work (*simulation* mis-specification), but also arises in many unsupervised methods (such as maximum-likelihood or Bayesian probabilistic models). In these cases, mis-specification typically results from simplified or incorrect assumptions built into a probabilistic model (*model* mis-specification, reviewed in detail by [56]). Such model mis-specification can be difficult and time-consuming to identify and address, usually calling for careful experimental design and model comparison [56]. In some ways, the simulation mis-specification problem is more straightforward to address through fully empirical, data-driven solutions such as domain adaptation. It remains to be seen whether these empirical techniques can be used to improve probabilistic-model-based inference methods. Overall, there is rich potential for new work to address a wide variety of mis-specification challenges in population genetics, leading to improved accuracy and robustness in inference.

## Methods

### Methodological summary of unsupervised domain adaptation

To build domain-adaptive versions of SIA and ReLERNN, we opted for the neural network architecture proposed by Ganin & Lempitsky [30], which involved attaching a domain classifier branch via a gradient reversal layer (GRL) to a layer of the original neural network where a latent representation of the data is presumably obtained. For example, in a CNN, the attachment point is usually immediately after the convolutional and pooling layers, which are primarily responsible for feature extraction. One possible heuristic for picking the attachment point is to look for a "bottleneck layer" in the original network corresponding to the lowest-dimensional representation of the input. The GRL-containing networks consist of three components–a label predictor branch, a domain classifier branch and a feature extractor common to both branches (**Fig 2A and 2B**). During the feedforward step, when data is fed to the neural network to obtain prediction outputs in both branches, the GRL is inactive; it simply passes along any input to the next layer. However, during backpropagation, when the gradient of the loss function with respect to the weights of the network is calculated iteratively backward from the output layer, the GRL inverts the sign of any incoming gradient before passing it back to the previous layer. This operation has the effect of driving the feature extractor away from distinguishing the source and target domains, and consequently encourages it to extract "domain-invariant" features of the data. This effect is manifested during training as the domain-classifier loss being *maximized*. We implemented the GRLs in TensorFlow (v2.4.1) using the 'tf.custom_gradient' decorator. On top of each custom GRL, the rest of the model was built using the 'tf.keras' functional API (see the GitHub repository for details).

All models were trained with the Adam optimizer using a batch size of 64. For the domain-adaptive models, training consisted of both (1) feeding labeled data from the source domain through the label predictor and obtaining a label prediction loss (cross entropy for classification task, mean squared error for regression task); and (2) feeding a mixture of unlabeled data from both the source and target domains through the domain classifier, obtaining a domain classification loss (cross entropy) (**Fig 2C**). In each minibatch, back-propagation from these two steps occurred simultaneously (i.e. the weights of the feature extractor were updated according to the combination of gradient from the label predictor and reversed gradient from the domain classifier). Note that the same source-domain data (but shuffled differently) were used for both steps. Training was accomplished using a custom data generator implemented with 'tf.keras.utils.Sequence'. In this study, we simply assigned equal weights to the label-prediction and domain-classification loss functions (following [30]). Nonetheless, the relative weights of the two branches can be tuned via a hyper-parameter $\lambda$, with potential implications for performance. Intuitively, the domain classifier should be penalized more when the simulations are more mis-specified. One potential strategy is to leverage the losses and gradients of the domain classifier to guide the choice of $\lambda$. Each training epoch took around 300 s for the domain-adaptive SIA model and around 800 s for the domain-adaptive ReLERNN model on a single NVIDIA Tesla V100 GPU. With early-stopping, the models in this study were trained on average for tens of epochs. The runtimes for domain-adaptive SIA and ReLERNN models were therefore on par with their standard versions (on the order of hours) [10,12].

### Setup of benchmarking experiments

We designed four benchmarking scenarios to contextualize the performance of the domain-adaptive models (**Fig 1C**). *i*) In the *simulation benchmark* (*source-matched*) case, we tested the original model trained with source domain data on held-out samples in the source domain.

This is how model benchmarks are usually run, with the test data following the same distribution as the training data. Note that for the SIA model, the source domain consists of true genealogies and therefore both training and testing were performed with true trees. *ii*) In the *hypothetical true model* (*target-matched*) case, the original model was trained and tested with labeled target domain data. Here, both training and testing were performed with inferred genealogies for the SIA model. This is a hypothetical case because it is unlikely in the evolution setting to have large quantities of labeled data from the target domain for training (i.e. real population data with known ground truth of evolutionary parameters). This case represents the performance ceiling of a standard machine learning model trained in-domain. *iii*) The *standard model application* recapitulated the usual workflow of supervised machine learning methods, where the model trained with source domain simulations was applied directly to "real" data in the target domain. This was the baseline case to which we compared the domain-adaptive model. *iv*) *Domain-adaptive application* of supervised machine learning models is the novel approach introduced in this study (see above and **Fig 1A**).

## Background selection experiment with SIA

To assess the robustness of domain-adaptive SIA (dadaSIA) to background selection, we simulated labeled examples (250,000 neutral and 250,000 sweep) in the source domain under demographic equilibrium with $N_e = 10,000$ and $\mu = \rho = 1.25\times10^{-8}$/bp/gen. The sweep simulations consisted of 100kb chromosomal segments with a hard sweep at the central nucleotide having selection coefficient $s \in [0.002, 0.01]$. Simulations were performed in SLiM 3 [15,16] followed by recapitation with msprime [17], and we kept the true genealogies as source domain data. The unlabeled data in the target domain (with the exception of held-out test dataset with labels retained) were simulated in a similar fashion, albeit with a 10kb segment ("gene") under purifying selection at the center of each 100kb chromosomal segment. All mutations in the central 10kb segment that arose during the forward stage of the simulations (in SLiM), other than the beneficial mutation in sweep simulations, followed a DFE parameterized by a gamma distribution with a mean $\bar{s} = -0.03$, a shape parameter $\alpha = 0.2$ and had dominance coefficient $h = 0.25$ [34]. We retained only the sequence data from the target domain simulations and inferred genealogies using Relate [57]. The datasets were partitioned following a 90%:2%:8% train-validation-test split.

## Demography mis-specification experiment with SIA

In a second set of simulations, we gauged whether domain adaptation also protects SIA against demographic mis-specification. In this case, instead of specifying the degree of mis-specification *a priori*, we designed an end-to-end workflow that recapitulated how demographic mis-specification arises in a realistic population genetic analysis (**S1A Fig**). First, we simulated "real" data (in the target domain) using an assumed demography (**S1A Fig**, loosely based on the three-population model in [58]). Similar to what one would do with actual sequence data, we then used the "real" samples to infer a demography with G-PhoCS [35], pretending that the true demography and genealogies were unknown. The G-PhoCS model assumed constant population sizes between split events and a single pulse migration from population C to B, and therefore was under-parameterized. As shown in **S1A Fig**, the inferred demography was consequently somewhat mis-specified. In addition to errors in population sizes, the split between B and C was inferred to be much more recent compared to the true demographic model. This mis-specified demographic model was then used to simulate labeled training data (in the source domain) for SIA.

 With the goal of using SIA to infer selection in population B, we simulated a soft sweep site at the center of a 100kb chromosomal segment with selection coefficient $s \in [0.003, 0.02]$ and

initial sweep frequency $f_{init} \in [0.01, 0.1]$, under positive selection only in population B. To improve computational efficiency, simulations were performed with a hybrid approach where the neutral demographic processes were simulated first with msprime [17], followed by positive selection simulated with SLiM 3 [15,16]. We produced 200,000 balanced (between neutral and sweep) simulations of "real" data, 10,000 of which were randomly held out as ground-truth test data for benchmarking with their labels preserved (**S1A Fig**). The rest remained unlabeled. This corresponded to a train-validation-test split of 93%:2%:5%. We preserved only the sequences and used Relate [57] to infer the ARG of population B from the "real" data. SIA works with a single population and thus the central genealogies containing only samples from population B were encoded as input to the model. For demographic inference, we randomly downsampled 10,000 5kb loci and analyzed them with G-PhoCS, keeping 4 (diploid) individuals from population A and 16 (diploid) individuals each from populations B and C. We took the median of 90,000 MCMC samples (after 10,000 burn-in iterations) as the inferred demography (shown in **S1A Fig**). The control file used to run G-PhoCS is available in the GitHub repository. We then simulated true genealogies of population B using the inferred demography, yielding 200,000 balanced samples with neutral/sweep and selection coefficient labels. All SIA models in this study used 64 diploid samples (128 taxa).

### Running SIA under varying degrees of simulation mis-specification

To probe the limit of domain adaptation in mitigating simulation mis-specification, we performed a series of experiments that gradually increased the severity of mis-specification. In all cases, the source domain consisted of 400,000 balanced samples of *true* genealogies simulated under a constant $N_e$ of 10,000. The target domain had a matching size of 400,000 balanced samples of *inferred* genealogies. We used $\mu = \rho = 1.25 \times 10^{-8}$/bp/gen unless otherwise specified. The datasets were partitioned following an 87.5%:2.5%:10% train-validation-test split. In the "tree inference only" case, the target domain consisted of *inferred* genealogies simulated under a constant $N_e$ of 10,000 with no demographic mis-specification. In addition, we tested four cases with $N_e$ = 8,000, 5,000, 2,000 or 500 bottlenecks between 1,000 and 2,000 generations before the present, respectively (**S4 Fig**). Finally, we tested an "extreme" case with the $N_e$ = 500 bottleneck, a mis-specified $\rho = 1 \times 10^{-7}$, as well as background selection in the central 10kb region following a DFE parameterized by a gamma distribution with a mean $\bar{s} = -0.03$, a shape parameter $\alpha = 0.2$ and a dominance coefficient $h = 0.25$.

### Updates to genealogical features and deep learning architecture for the SIA model

For this study, we adopted a richer encoding of genealogies than the one used previously for SIA. Instead of simply counting the lineages remaining in the genealogy at discrete time points [12], we fully encoded the topology and branch lengths of the tree using the scheme introduced by [59]. Under this scheme, a genealogy with $n$ taxa is uniquely encoded by an $(n\text{-}1) \times (n\text{-}1)$ lower-triangular matrix $F$ and a weight matrix $W$ of the same shape. Each cell $(i, j)$ of $F$ records the lineage count between coalescent times $t_{n-j}$ and $t_{n-1-i}$, whereas each cell $(i, j)$ of $W$ records the corresponding interval between coalescent times, $t_{n-j} - t_{n-1-i}$ (see **S1B Fig** and [59] for details). In addition, we used a third matrix $R$ to identify the subtree carrying the derived alleles at the site of interest, following the same logic as $F$ (see **S1B Fig** for an example). The $F$, $W$ and $R$ matrices have the same shape and therefore can easily be stacked as input to a convolutional layer with three channels (**Fig 2A**, 128 taxa yield a 127 x 127 x 3 input tensor).

Unlike the previous reductive encoding of lineage counts, the new scheme is bijective [59] and therefore contains the entirety of information in the genealogy. To utilize the improved

input feature consisting of stacks of matrices, we modified the neural network architecture of SIA and used convolutional layers (**Fig 2A**). The new feature encoding and convolutional neural network (CNN) architecture resulted in modest gain in performance compared to the original encoding and recurrent neural network (RNN) architecture (**S1C Fig**). In this study, both the standard and domain-adaptive SIA models use convolutional layers with the improved feature encoding. The original SIA codebase (github.com/CshlSiepelLab/arg-selection) has been updated to take advantage of the new feature encoding and model architecture as well.

## Simulation study of recombination rate inference with ReLERNN

We conducted two sets of simulation experiments to test the same two types of mis-specification as previously described for SIA. Each simulation consisted of 32 haploid samples of 300kb genomic segment with uniformly sampled mutation rate $\mu \sim U[1.875 \times 10^{-8}, 3.125 \times 10^{-8}]$ and recombination rate $\rho \sim U[0, 6.25 \times 10^{-8}]$. To test the effect of background selection, the labeled source domain data (with true values of $\rho$) were simulated under demographic equilibrium with $N_e$ = 10,000, whereas the unlabeled target domain data were simulated under the same demography, but with the central 100kb region under purifying selection, as with SIA. To test the effect of demographic mis-specification, we conducted simulations similar to those of [10] where labeled source domain data were generated under demographic equilibrium (with $N_e$ = 6,000, calculated approximately by $\frac{\hat{\theta}_W}{4\mu}$ where $\hat{\theta}_W$ was estimated from the target domain data) and unlabeled target domain data were generated under a European demography [36]. For each domain, 500,000 simulations were generated with SLiM 3 (background selection experiment) or msprime (demography experiment), and partitioned following an 88%:2%:10% train-validation-test composition. We modified the ReLERNN model to be domain-adaptive (**Fig 2B**) and used the simulated data to benchmark its performance against the original version of the model.

## Application of domain-adaptive SIA model to 1000 Genomes CEU population

Labeled training data (source domain) for SIA were simulated with discoal [60] under the European demographic model from [36]. Following [12], we simulated 500,000 100-kb regions of 198 haploid sequences. The per-base per-generation mutation rate ($\mu$) and recombination rate ($\rho$) of each simulation were sampled uniformly from the interval $[1.25 \times 10^{-8}, 2.5 \times 10^{-8}]$; the segregating frequency of the beneficial allele ($f$) was sampled uniformly from [0.05, 0.95]; the selection coefficient ($s$) was sampled from an equal mixture of a uniform and a log-uniform distribution with the support $[1 \times 10^{-4}, 2 \times 10^{-2}]$. An additional 500,000 neutral regions were simulated to train the classification model, under the identical setup sans the positively selected site.

We curated target domain data from the 1000 Genomes CEU population to train the domain-adaptive SIA model (dadaSIA). The genome was first divided into 2Mb windows 1,111 of which passed three data-quality filters: 1) contained at least 5,000 variants, 2) at least 80% of these variants had ancestral allele information, and 3) at least 60% of nucleotide sites in the window passed *both* the 1000 Genomes strict accessibility mask [1] and the deCODE recombination hotspot mask (standardized recombination rate > 10; [61]). In each of these 1,111 windows, we randomly sampled 1,000 variants and extracted genealogical features at those variants from Relate-inferred ARGs [57], yielding around 1 million samples that constituted the unlabeled target domain data. Finally, domain-adaptive SIA models for classifying sweeps and inferring selection coefficients were trained as described previously and applied to a collection of loci of interest (**Table 1**).

## Supporting information

**S1 Fig. Domain-adaptive SIA. A)** The workflow of a simulation study that aims to benchmark the performance of the domain-adaptive SIA model in a realistic setting of demographic mis-specification. **B)** An improved version of SIA input features that encodes the full genealogy (adapted from [59]). A genealogy with $n$ taxa at a polymorphic site is uniquely encoded by three ($n$-1) x ($n$-1) lower triangular matrices. The weight matrix **W** encodes the coalescent intervals where $w_{ij} = t_{n-j} - t_{n-1-i}, \forall i \geq j$, and the topology matrix **F** encodes the number of lineages persistent in the coalescent intervals *corresponding* to **W** (i.e. $f_{ij}$ = *# of lineages between* $t_{n-j}$ *and* $t_{n-1-i}, \forall i \geq j$). The de**r**ived lineage matrix **R** encodes only the subtree subtending the branch where the mutation occurred (red lightning symbol), following the same scheme as **F**. Note that the **W** matrix is a redundant encoding of the $n$-1 coalescent times ($t_1, t_2, \ldots, t_{n-1}$), which contains information roughly equivalent to the original SIA input features [12]. **C)** Comparison of the performance of the new SIA input features in (**B**) to that of the original SIA input features.
(TIF)

**S2 Fig. Selection coefficient inference performance of SIA models.** Raw data used to plot **Fig 3B** and **3D** are presented in (**A**) and (**B**), respectively. Performance of SIA models in the simulation experiment of failure to account for background selection (**C**) and in the simulation experiment of demographic model mis-specification (**D**) is presented in terms of mean and standard deviation of the absolute error (top) as well as the distribution of raw error (bottom). Statistical significance (\*) of the difference between the absolute error of the standard model and that of the domain-adaptive model is evaluated with Welch's *t*-test. See **Fig 1C** for definition of the model labels.
(TIF)

**S3 Fig. Performance of dadaSIA models trained with imbalanced data.** The sweep classification performance of dadaSIA models trained with different proportions of sweep vs. neutral examples in the target domain is shown in the form of precision-recall curves (**A**) and the area under precision-recall curve (AUPRC) (**B**). Note that the performance is always evaluated on a balanced test set. The performance of dadaSIA models trained with less target domain data than source domain data is shown in the form of precision-recall curves (**C**) and the values of AUPRC (**D**) for the classification task, and in the form of root mean squared error (RMSE) (**E**) for the selection coefficient inference task. The dashed lines in (**B**), (**D**) and (**E**) indicate performance of the standard model.
(TIF)

**S4 Fig. Demographic mis-specification in the form of different degrees of bottlenecks tested in Fig 5 experiments.**
(TIF)

**S5 Fig. Inference of out-of-range selection coefficients in the target domain using the dadaSIA model.** The dadaSIA model trained with source domain data under $s \in [0.01, 0.02]$ failed to meaningfully infer any value lower than 0.01, even when examples of $s \in [0.001, 0.01]$ were supplied to the model as "unlabeled" target domain data, and vice versa.
(TIF)

**S6 Fig. Recombination rate inference performance of ReLERNN models.** Raw data used to plot **Fig 4A** and **4B** are presented in (**A**) and (**B**), respectively. Performance of ReLERNN models in the simulation experiment of failure to account for background selection (**C**) and in the simulation experiment of demographic model mis-specification (**D**) is presented in terms of

mean and standard deviation of the absolute error (top) as well as the distribution of raw error (bottom). Statistical significance (*) of the difference between the absolute error of the standard model and that of the domain-adaptive model is evaluated with Welch's *t*-test. See **Fig 1C** for definition of the model labels.
(TIF)

**S7 Fig. Distribution of raw error of the ReLERNN models inferring recombination rate without simulation mis-specification.** The respective mean absolute error (MAE) of the standard and domain-adaptive models are $4.05 \times 10^{-9}$ and $4.13 \times 10^{-9}$, under demography equilibrium, and $4.28 \times 10^{-9}$ and $3.93 \times 10^{-9}$, under a European demography. Note that the domain-adaptive model has a slight upward bias in its estimates in the case of European demography.
(TIF)

**S8 Fig. Validation loss of the label predictor branch (mean squared error) and the domain classifier branch (binary cross entropy) over training epochs.** The losses of the domain-adaptive ReLERNN models during training are plotted with and without simulation mis-specification. The red dot marks the early-stopping epoch (i.e. epoch with the lowest validation loss for the label predictor).
(TIF)

**S9 Fig. Domain classifier loss of dadaSIA models under different degrees of simulation mis-specification.** See **Fig 5** and **Methods** for details of the types of mis-specification.
(TIF)

## Acknowledgments

We would like to thank Jesse Gillis, Peter Koo, David McCandlish, Armin Scheben, and Xander Xue for useful discussion.

## Author Contributions

**Conceptualization:** Ziyi Mo, Adam Siepel.

**Data curation:** Ziyi Mo.

**Formal analysis:** Ziyi Mo.

**Funding acquisition:** Adam Siepel.

**Investigation:** Ziyi Mo.

**Methodology:** Ziyi Mo.

**Resources:** Adam Siepel.

**Software:** Ziyi Mo.

**Supervision:** Adam Siepel.

**Validation:** Ziyi Mo.

**Visualization:** Ziyi Mo.

**Writing – original draft:** Ziyi Mo, Adam Siepel.

**Writing – review & editing:** Ziyi Mo, Adam Siepel.

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
