## [Decision Letter · Decision Letter 0]

11 May 2023

Dear Dr Siepel,

Thank you very much for submitting your Research Article entitled 'Domain-adaptive neural networks improve supervised machine learning based on simulated population genetic data' to PLOS Genetics.

The manuscript was fully evaluated at the editorial level and by independent peer reviewers. The reviewers appreciated the attention to an important problem, but raised some substantial concerns about the current manuscript. Based on the reviews, we will not be able to accept this version of the manuscript, but we would be willing to review a much-revised version. We cannot, of course, promise publication at that time.

Three reviewers have each given thorough and insightful critiques of this manuscript. I have also read this paper carefully and discussed it with many colleagues, and I am in close agreement with the reviewers' recommendations. I think this paper represents a potentially important step forward in the design and application of deep learning methods for population genetic inference, but additional analyses recommended by the reviewers would strengthen the paper considerably. I think the reviewers have made some excellent suggestions and I would like to see the authors address them in a revision.

I particular, I would like to stress the following points:

1) Reviewer #3 raises an important point that I think the authors should clarify: is misspecification a problem unique to machine learning methods, or a more universal problem in population genetic inference that machine learning methods may be especially useful for mitigating? I of course lean towards the second one here, and it is fine if the authors don't agree, but I would like them to share their thoughts on this.

2) I agree with the reviewers that pushing the misspecification boundary until things break would be a great, and probably essential, addition to this paper. Does domain adaptation ever hurt performance if the degree of misspecification gets to be too large (or too small)?

3) As noted by the reviewers, the correct interpretation of the authors' comparison to previous selection coefficients is unclear. Why should we expect that the previous non-SIA estimates would be unaffected by model misspecification? If I recall, some of these estimates may have been based on partial allele frequency trajectories inferred from ancient DNA, but even estimates of s obtained from population genetic time-series data will be biased by misspecification. I therefore urge the authors to clarify what they think the take-home from this comparison is.

The additional points/suggestions raised by the reviewers are also important, and should also be addressed in a revision.

If you decide to revise the manuscript for further consideration at PLOS Genetics, please aim to resubmit within the next 60 days, unless it will take extra time to address the concerns of the reviewers, in which case we would appreciate an expected resubmission date by email to plosgenetics@plos.org.

Please do not hesitate to contact us if you have any concerns or questions.

Yours sincerely,

Daniel R Schrider

Guest Editor

PLOS Genetics

Gregory P. Copenhaver

Editor-in-Chief

PLOS Genetics

Reviewer's Responses to Questions

**Comments to the Authors:**

Reviewer #1: This study presents a careful application of domain adaptation techniques from comp sci to some example analyses in population genetics and provides a well written explanation of the approach. I view this as an important paper that I expect may benefit the field of pop gen, and it was very thought provoking. The main idea I took away, is the domain classifier branch with GRL *can only help* and should maybe be added to every neural network architecture; because, every empirical dataset is misspecified relative to the training set, and we don't know the full extent or quality of misspecification or else we could just simulate data that looks real (e.g. that includes background selection, accurate demography, etc). My biggest concern is whether, in some scenarios, tacking on the domain classifier branch to a network may actually hurt our inferences; this is important for guiding when and where to use domain adaptation. I reiterate this concern in the below list of comments and suggestions.

L75. "no way to ensure that the ranges considered are adequately large". I kept an eye out to see if you addressed "out of prior range" scenarios, but I don't think this was explored. Your misspecification scenarios were "qualitative" in that the training data lacked a piece of biological realism that was present in the test data. I'm curious about a more "quantitative" misspecification, where you try to estimate selection coefficients outside of the range shown in Figure 2. Or, for classifying sweeps, use test data that have very small or very large population sizes. I don't view this as totally necessary for your already useful analysis, but it seems like a potentially high reward simulation experiment.

I'm wondering about how to implement domain adaptation for other networks. (i) I was looking for explanation about how the "insertion point" is decided for the GRL/domain classifier branch? It seems to be an intermediate point in the network, but I can imagine this looking quite different for some other networks. (ii) How much target domain data is sufficient? (iii) The 20/80 experiment (and other proportions of sweep/no sweep) for the target domain was a great idea. In cases where we have some control over the target domain, e.g. exonic versus non-exonic regions, synonymous versus nonsynonymous snps, population 1 versus population 2—is it always better to balance the target domain?

The 20/80 experiment was very useful. It did stand out to me that the 100% sweep case performed worse than the standard model. Although we shouldn't worry about 100% of human loci representing sweeps as in this particular experiment, I think this result is important to highlight in the main text, because it shows that domain adaptation is not necessarily >= the standard model. It makes me wonder what other weird test sets might cause the domain adapted model to perform not as well, and whether *my* empirical data are approprite to use with domain adaptation.

Your simulation experiments are very convincing and cleanly implemented. I have tried hard to think of possible exceptions that might break things. I would expect the domain classifier to have no effect if the target domain was very similar to the source domain; but, is there any chance that training would suffer? I'm sort of thinking of GANs, how if the discriminator does poorly at the onset then the quality of training decreases. More likely, would it decrease prediction accuracy of the trained model slightly if the extra domain classifier branch was added, but not needed? Related, is there any chance that the added domain classifier branch of the network could be helping the network along simply by including additional layers and model parameters? Similarly, could the network be benefitting from showing it extra data, or the same source domain data twice, in a way that doesn't help it adapt the domain, but simply represents a bigger training set?

In the ReLERNN analysis, the GRL led to overcorrected predictions it seems. I thought this was an interesting "side effect" that might be discussed. Do you think that tuning the lambda parameter might alleviate this (although it seems hard to tune this with real data)?

L254. In the empirical results section, "these observations suggest that the addition of domain adaptation does not radically alter SIA’s predictions for real data but may in some cases improve them" (1) does not acknolwedge the possibility that the results are worse than before, and (2) could be interpreted as "domain adaptation DID improve some estimates", but as you point out, we don't know the truth with these real data. And, the fact that the new results are dragged slightly towards the mean from the other methods that presumably do not account for background selection or demography does not support that the new appoach is performing better, right? Unless you assume that Standard SIA performs poorly relative to the other methods? L309. Likewise, it doesn't seem like the evidence is stong enough here, unless you knew that the other methods are more accurate.

I think it's perhaps interesting, reassuring in fact, that the results aren't so much different than Standard SIA in the empirical analysis. I was concerned however that SIA doesn't seem very accurate in Figure 2 even with Domain Adaptation. It seems to learn *some* signal for selection coefficient, but overall the predictions are quite noisy. If additional empirical analyses were performed I'd be curious to see maybe how the ReLeaRN results were affected. Like Relearn with and without domain adaptation applied to some real data (the paper did Drosophila I think?)

I realize that the specific domain adaptation technique you use was described previously, but the technical description, and intuition behind the approach, seems like they could use additional explanation.

In your experiments, the "real" data are relatively similar to the "fake" training data, in the sense that they are both simulated with relatively simple models. However I was left wondering about the limits of domain adaptation. Real world empirical data have more than just background selection separating them from the training set, but also various other kinds of selection, intricate demographic histories (I know you looked at demography separately), population structure, and many other subtle factors that shape genetic variation. I think it's important to explore more complex scenarios, in order to answer the question: is there any potential downside to tacking the domain classifier, in practice? Does it ever contribute to additional error, if say, the real data are very very different from the simulated data. I suggest following up with an even more complicated misspecification scenario. For example one that includes BGS and demographic history simultaneously, the empirical ARG inference step as before, plus misspecified mutation rate, recombination rate, and gene conversion? Since releRNN seems to perform really well on the current simulations, perhaps they can be badly misspecified and some signal will still be retained. It might perform quite well on the badly misspecified data with the addition of domain adaptation! But ideally, I think, things should be pushed to the limit to understand how this works. Could also calculate summary statistics for each test set to quantify the divergence between source and target domains. I view this "badly misspecified" experiment as an important piece that will help guide when and if to use domain adaptation.

Minor points:

Fig 1A. It could just be me, but I was unfamiliar with color-coded DNA alignment in the top left, didn't know what the colors meant.

I thought that SIA was an RNN, so I was confused about the conv/pool layers in figure 2 of the current paper?

L88. I agree that ABC methods are costly in the sense that extreme numbers of simulations are required to explore such vast parameter space; however there are nice approaches that deal with the many summary statistics, like the neural network regression from Blum and Francois (2010) and ABC with random forests (Pudlo et al 2016, Raynal et al 2019).

L102. Is domain adaptation synonymous with generalization? I'm not certain but I thought the framing here might be "A variety of strategies for generalization have been introduced ... The specific kind we're interested in is domain adaptation".

L104. "domain adaptation can be incorporated directly into the process of training". It seems like the proceeding approaches also are incorporated in the training process. Maybe domain adaptation is distinct because it shows target-domain data to the network.

I for one could have used a brief description of precision-recall curves and the AUC metric. Maybe not necessary.

Nothing is wrong with the contour plots (except, the dashed line is orange instead of yellow), but I personally like the scatter plots in the supp mat a whole lot better. Especially for the ReLeRNN results.

L142. "reversing the gradient" I thought could use a bit more explanation at this stage, although I see you guide readers to the methods section. Maybe "reversing the sign of the gradient of the loss for the domain classifier for layers in the feature extractor"? Maybe not necessary, but it would have helped me if the architecture was explained more up front (and more clearly, see my below comments).

L175. talks about the importance of the ARG inference step. It would have helped me follow along if a "(See methods section)" was added, here. And also if the ARG inference step was mentioned in the figure 3 caption it would be helpful I think.

L201. Mentions an alternative demographic model. But at this point, we don't know what the first demographic model used for SIA was.

L248. Should "In all other cases" be "In all cases"?

L327. Confusing whether "GRL" refers to just one layer, or all layers in the domain classifier branch. I think it's just one layer, so it seems like this first sentence should instead say "we added a domain classifier branch with a GRL to the network." Seems like the domain classifier branch as a whole is insufficiently explained.

L332. It's unclear if "output layer" refers to the output in the label predictor branch, the output in the domain classifier branch, or both?

L380. peudo-real

L439. 1,111 windows... were 1000 SNPs sampled many times? How did it turn into 1,000,000 samples?

I would assume it's correct, but for the ASIP gene in the empirical analysis the previous estimate is 0.09 which seemed high.

Reviewer #2: In this paper the authors present domain adaptive versions of two population genetics methods: SIA and ReLERNN. Since both methods (as well as most machine learning methods for population genetics) use simulated data for training, mismatches between training and testing/real data can cause issues in accuracy and interpretation of results. By incorporating a GRL (gradient reversal layer) component into the neural network architectures, the authors are able to mitigate the mismatch between the source (training) and target (testing) domains. The GRL part of the network is trained to identify source vs target examples, but then the gradient values wrt the weights are reversed, encouraging the overall network to use features common to both the source and target domains.

The authors apply both methods to "real" data simulated under different models from the training data, demonstrating improved inference results in the domain-adaptive versions. dadaSIA is also applied to real data (from European population CEU), although the results are a bit difficult to interpret.

Overall I love the ideas in this paper - the methodological advances could be widely applicable for other population genetics methods and could help us as field understand what features of simulated training data are important for different applications. I have a few concerns with some of the details and a few suggestions for improving the manuscript.

Major comments:------------------

1) The "simulation benchmark" (source-matched) and "hypothetical true model" (target-matched) experiments need to be described more clearly early on (starting around Line 163). I think this distinction is confusing to readers because when both the source and target datasets are simulated, the difference between "source-matched" and "target-matched" is not very meaningful for most methods (as you mention in Line 294). SIA is more of a special case where ARG inference is needed.

This is not a strong suggestion (as I would be open to reasons why not), but I would lean toward removing the "simulation benchmark" and just include the target-matched benchmark (as this is the desired case when the simulations match the "real" data). Further, I would perhaps suggest that the "standard" SIA model *not* include the true ARGs, but instead infer them in the same fashion as the real data. It feels unsatisfying to throw that information away from the simulations, but not doing so heightens the gap between the treatment of real and simulated data and makes it difficult to separate if the domain-adaptation modification is picking up on ARG inference issues vs. other differences between real/simulated data. Alternatively, you could try both (as is done in the original SIA paper I believe?)

2) SIA data representation. The ARG representation is quite different in this paper vs the original SIA paper. Was this choice related to the domain-adaptive version or just a more complete ARG representation? Does it improve results to use this representation in the standard model case? Overall I think this representation difference should be made a bit more clear throughout - as readers looking back at the original SIA paper may not be able to compare the results directly.

3) CEU selection results. I think these should be reframed a bit (still included though). It is possible the dadaSIA results represents an improvement over SIA, but the number of examples is low and I am more struck by the differences between both SIA approaches and other literature. It would be interesting to investigate the causes of these differences (maybe more to do with general method differences?), perhaps in a future study.

On this section - were the previously reported selection coefficients (from SIA) using the SIA method in [12] or the different ARG representation in this paper? If from [12] then it would be difficult to separate the difference due to the representation change vs the domain-adaptation modification, right?

4) It seems like the demographic mis-specification for SIA (infer the demography from simulated "real" data) is not as extreme as for ReLERNN (where the source data has a constant population size and the target does not). Along with the ARG inference component, I think this makes it difficult to compare the results. The improvement for ReLERNN is not as large as for SIA (despite the bigger difference in source vs target). The authors highlight differences between SIA and ReLERNN in the Discussion, but I think it might be worth caveating the ReLERNN results a bit more - perhaps this method is more robust to simulation mis-specification than most.

5) Interpretability. I think it would be very interesting to understand what the GRL is learning. Perhaps outside the scope of this study, but maybe the authors could include some discussion about how we might better understand gaps between simulated/real data through interpreting this component.

Minor comments:------------------

-Line 59: "efficiently produce extremely large (virtually unlimited)" caveat this a bit... some selection simulations with large population sizes still very slow

-Line 89: "scale poorly with the number of summary statistics" the bigger issue is (usually) that the simulations scale poorly with the number of *parameters* in the model. If summary stats are quadratic in the number of samples or number of sites, these can scale poorly too, but it's not usually the number of summary stats that's the problem.

-Line 103: "reweighting training instances" could you go into a bit more detail?

-Line 150: "the demographic model used for the source domain was estimated from "real" data generated under a more complex demographic model and was therefore somewhat mis-specified"

From Fig S1A it seems like it's the same demographic model structure (just with inferred parameters) for SIA - maybe I missed something? (For ReLERNN it's more complex though.)

Methods: Could you include the # of trainable params/weights in each component?

Line 341: For training, is it the same source data for both components? I didn't quite understand if training is iterative (i.e. mixed source/target batch, update weights, then source batch, update weights etc) or not

Line 374: Maybe I missed this, but could you explain population A, B, C? This is from the out-of-Africa model right?

Background selection section in Methods: (Line 357) "All mutations..." isn't there also a positively selected mutation for the "selection" examples?

Editing:------------------

-Line 42: "the European" -> "a European"

-Line 123: missing "a"

-Line 173: "on it" -> rephrase

-Line 270: "ignored" -> "un-modeled"

Reviewer #3: attached

**Have all data underlying the figures and results presented in the manuscript been provided?**

Reviewer #1: None

Reviewer #2: Yes

Reviewer #3: Yes

PLOS authors have the option to publish the peer review history of their article (what does this mean?). If published, this will include your full peer review and any attached files.

Reviewer #1: No

Reviewer #2: No

Reviewer #3: No

---

## [Decision Letter · Decision Letter 1]

18 Oct 2023

Dear Dr Siepel,

Thank you very much for submitting your Research Article entitled 'Domain-adaptive neural networks improve supervised machine learning based on simulated population genetic data' to PLOS Genetics.

The manuscript was fully evaluated at the editorial level and by independent peer reviewers. The reviewers appreciated the attention to an important topic but identified some concerns that we ask you address in a revised manuscript.

We therefore ask you to modify the manuscript according to the review recommendations. Your revisions should address the specific points made by each reviewer.

Yours sincerely,

Daniel R Schrider

Guest Editor

PLOS Genetics

Gregory P. Copenhaver

Editor-in-Chief

PLOS Genetics

The reviewers and I were generally pleased with the revisions and are enthusiastic about the paper. There are just a few outstanding minor comments where some additional change/clarification in the text is necessary.

Reviewer's Responses to Questions

**Comments to the Authors:**

Reviewer #1: The authors sufficiently addressed all of my comments and questions. They also did substantial simulation experiments exploring the limits of the technique, which I appreciate, and the revised presentation seems effective and fair. I recommending accepting this great paper.

Reviewer #2: The authors have completed a substantial revision of their original paper. In particular the new analysis of varying levels of mis-specification is very helpful, along with the inclusion of loss curves for the different components of the network. They have also clarified the real data CEU results and generally improved the manuscript to make many aspects clearer. My initial comments have all been addressed, including the differences (in data encoding and architecture) between SIA and dadaSIA which are clear and consistent now. I just have a few questions about the loss analysis, in particular this interesting observation:

"We observed that the domain classifier generally tended to start with a lower loss and took more epochs to train when the mis-specification is more severe (Fig. S9). It might be worthwhile, as a future endeavor, to try to identify the features driving this loss, understand their evolutionary significance, and, perhaps, incorporate them into a new set of simulations. In such a way, domain adaptation could be used to discover evolutionary processes and improve the models used for simulation."

The idea of a lower loss when there is more mis-specification make sense to me. But when I was thinking about the domain-classifier loss trajectory (and also went back to the original domain-adaptive paper [30]) it seems like we are still trying to *minimize* the domain-classifier loss (i.e. learn the difference between the source and target domains) right? But due to gradient-reversal in the feature-extractor, this becomes difficult over training, so the loss increases? I guess I expected to see that as mis-specification increases, the loss would be somewhat low even at the end of training (i.e. still easy to distinguish source and target). I was surprised to see that in Fig S9 for example, all models eventually achieved random guessing (albeit with the "extreme" case happening more slowly). I was also surprised that the loss was almost 0 (i.e. no classification errors) right away. Maybe for the more extreme mis-specifications it's an easy classification problem, but it seems like it would take at least a bit of time to learn the difference.

I think I'm missing something and that's leading to my confusion - if the authors could clarify the loss results a bit more that would be very helpful!

Reviewer #3: I am very positive about the revision of the manuscript. The authors made a substantial amount of experiments to address my comments and clarified the unclear sections. I read with interest the outcome of these new experiments and I am happy to see that they confirm the intuitions of the authors, reviewers and editor. It is a good news that the domain adaptation does not decrease the performance where there is little or no mis-specification, i.e. that the classifier was not weirdly overfitting. It is also a good news that even with little target data (1:10 of the source size) the performance remain similar or better than the standard model, making me more confident that DA is suitable in practice. Finally, I was glad to see that losses gave some insights on mis-specification levels and on the inner working of the network. Overall, I am very happy with the authors’ work and, as I had already said, convince of the study relevance.

Minor comments:

l.458 I would not classify ABC as an unsupervised method since the simulation set has labels that are used for inferring the posterior. ABC with correction steps are even learning a supervised machine learning model (penalized regression, dense neural network, ..) locally (in the region of the observed data)

- The authors said they were not sure what I was referring to in one of my comment

Me: Was this phenomenon of increased confidence observed in the previous experiments?

Them: “We are not sure what “previous experiments” refer to.”

What I meant is that the experiments to benchmark the dadaSIA are commented in terms of RMSE, AUPRC etc. Not in terms of prediction confidence. The authors observed and increase of confidence of dadaSIA compared to SIA for real data application (l.308, 403). Did they also observe a change in confidence between SIA and dadaSIA when applied to simulations?

It might be another advantage of domain adaptation if confidence appears to be well calibrated. (But can be kept for future study if the answer is not obvious).

**Have all data underlying the figures and results presented in the manuscript been provided?**

Reviewer #1: None

Reviewer #2: Yes

Reviewer #3: Yes

PLOS authors have the option to publish the peer review history of their article (what does this mean?). If published, this will include your full peer review and any attached files.

Reviewer #1: No

Reviewer #2: No

Reviewer #3: No

---

## [Editor Report · Decision Letter 2]

23 Oct 2023

Dear Dr Siepel,

We are pleased to inform you that your manuscript entitled "Domain-adaptive neural networks improve supervised machine learning based on simulated population genetic data" has been editorially accepted for publication in PLOS Genetics. Congratulations!

Yours sincerely,

Daniel R Schrider

Guest Editor

PLOS Genetics

Gregory P. Copenhaver

Editor-in-Chief

PLOS Genetics

Comments from the reviewers (if applicable):

**Data Deposition**

http://datadryad.org/submit?journalID=pgenetics&manu=PGENETICS-D-23-00305R2

**Press Queries**

---

## [Editor Report · Acceptance letter]

31 Oct 2023

PGENETICS-D-23-00305R2 

Domain-adaptive neural networks improve supervised machine learning based on simulated population genetic data 

Dear Dr Siepel, 

We are pleased to inform you that your manuscript entitled "Domain-adaptive neural networks improve supervised machine learning based on simulated population genetic data" has been formally accepted for publication in PLOS Genetics! Your manuscript is now with our production department and you will be notified of the publication date in due course.

With kind regards,

Anita Estes

PLOS Genetics

On behalf of:
